

# PLAUR facilitates the progression of clear cell renal cell carcinoma by activating the PI3K/AKT/mTOR signaling pathway

Tianzi Qin[1,2], Minyu Huang[2], Wenjuan Wei[3], Wei Zhou[2], Qianli Tang[1,4], Qun Huang[2], Ning Tang[5] and Shasha Gai[5]

[1] The First Clinical Medical College of Jinan University, Guangzhou, China
[2] Department of Urology, the Affiliated Hospital of Youjinag Medical University for Nationalities, Baise, China
[3] Department of Ultrasound department, the Affiliated Hospital of Youjinag Medical University for Nationalities, Baise, China
[4] The Affiliated Hospital of Youjinag Medical University for Nationalities, Baise, China
[5] Youjinag Medical University for Nationalities, Baise, China

Corresponding author
Qianli Tang, htmgx@163.com

## ABSTRACT

**Background**. PLAUR has been found upregulated in various tumors and closely correlated with the malignant phenotype of tumor cells. The aim of this study was to investigate the relationship between PLAUR and clear cell renal cell carcinoma (ccRCC) and its potential mechanism of promoting tumor progression.

**Methods**. The expression levels and clinical significance of PLAUR, along with the associated signaling pathways, were extensively investigated in ccRCC samples obtained from The Cancer Genome Atlas (TCGA). PLAUR expression in 20 pairs of ccRCC tumor tissues and the adjacent tissues was assessed using qRT-PCR and IHC staining. Additionally, a series of *in vitro* experiments were conducted to investigate the impact of PLAUR suppression on cellular proliferation, migration, invasion, cell cycle progression, and apoptosis in ccRCC. The Western blot analysis was employed to investigate the expression levels of pivotal genes associated with the PI3K/AKT/mTOR signaling pathway.

**Results**. The expression of PLAUR was significantly upregulated in ccRCC compared to normal renal tissues, and higher PLAUR expression in ccRCC was associated with a poorer prognosis than low expression. The *in-vitro* functional investigations demonstrated that knockdown of PLAUR significantly attenuated the proliferation, migration, and invasion capabilities of ccRCC cells. Concurrently, PLAUR knockdown effectively induced cellular apoptosis, modulated the cell cycle, inhibited the EMT process, and attenuated the activation of the PI3K/AKT/mTOR signaling pathway. PLAUR may represent a key mechanism underlying ccRCC progression.

**Conclusions**. The involvement of PLAUR in ccRCC progression may be achieved through the activation of the PI3K/AKT/mTOR signaling pathway, making it a reliable biomarker for the identification and prediction of ccRCC.

## INTRODUCTION

The mortality rate of renal cell carcinoma (RCC) is the highest among all urinary tract malignancies, accounting for approximately 2%–3% of all adult malignancies (*Yong, Stewart & Frezza, 2020*). Clear cell renal cell carcinoma (ccRCC) is the most prevalent form among the various pathological forms of RCC (*Siegel et al., 2023*). ccRCC has been extensively investigated, revealing a robust association with an unfavorable prognosis subsequent to metastasis (*Rodrigues et al., 2018*). Since its clinical symptoms in early stages are relatively insidious, patients present with clinical manifestations such as obvious hemabdominal mass only when the disease has already progressed to intermediate or advanced stages (*Yarandi & Shirali, 2023*). Unfortunately, at this stage, surgical treatment is no longer feasible for most patients (*Choi et al., 2017*). Moreover, ccRCC exhibits a relatively unfavorable outcome upon metastasis with an overall survival (OS) rate of <20% at 5 years (*Fendler et al., 2020*). Given the high propensity for recurrence, invasion, and metastasis in late-stage ccRCC, achieving a higher OS rate is extremely challenging (*Zhao et al., 2019*), Consequently, preventing cancer invasion and metastasis has emerged as a pivotal approach for ccRCC treatment. Despite certain positive outcomes with advancements in molecular targeted drugs and immunotherapy, long-term efficacy benefits are still limited (*Mattila, Vainio & Jaakkola, 2022*). Hence, it is imperative to identify novel molecular targets that can serve as indicators of ccRCC treatment or progression, as well as simultaneously elucidate the underlying molecular mechanisms driving ccRCC pathogenesis. More recently, as revealed by the comprehensive pan-cancer analysis results of the entire ccRCC genome, there are strong associations between specific mutations in ccRCC tumors and tumor development dynamics, RNA expression, and immunologic characteristics (*Zhang et al., 2020*). This discovery provides novel insights into tumor diagnosis and treatment.

In recent years, as the molecular understanding of cancer advances, the concept of precision medicine is gaining attention, that is, molecular profiles of tumors to guide therapeutic decision-making (*Wang et al., 2018*). The adhesion between tumor cells and endothelial cells is an early event in tumorigenesis, wherein the fibrolytic system functions. Consequently, dysregulation of this system may lead to the progression of multiple malignant solid tumors (*Mattila, Vainio & Jaakkola, 2022*). The fibrinolytic system, comprising plasmin (Pm) and its precursor plasminogen (Pg), is regulated by two endogenous activators referred to as tissue plasminogen activator (tPA) and urokinase-type plasminogen activator (uPA). Additionally, there exists a receptor urokinase type plasminogen activator receptor (uPAR or PLAUR), which binds with both tPA and plasminogen as annexin II on cell surfaces (*Kwaan, 2022*). The members of the fibrinolytic system play a pivotal role in facilitating interactions between cells and the extracellular matrix. This mechanism influences various cancer-related processes, including cellular differentiation, proliferation, metastasis, and angiogenesis. Among these members, PLAUR plays a pivotal role in cancer biology (*Zhang et al., 2020*). The PLAUR gene, located on chromosome 19q13, encodes the receptor for urokinase plasminogen activator (*Børglum et al., 1992*). It makes great contributions to facilitating the localization and advancement of plasminogen generation, and participates in various physiological and pathological

mechanisms of cell surface activation of plasminogen and localized breakdown of extracellular matrix (*Chen et al., 2022*). Additionally, it facilitates the binding of proprotein of urokinase plasminogen activator, which is receptor-binding proenzyme that can regulate proteolytic activity on tumor cell surfaces. Moreover, it participates in signaling pathways while regulating cell adhesion and migration (*Wang et al., 2018*). As has been evidenced previously, PLAUR expression is remarkably upregulated in various tumor cells, while being nearly undetectable in normal tissues; moreover, it exhibits a strong correlation with the malignant phenotype of tumor cells (*Børglum et al., 1992*; *Chen et al., 2022*; *Kwaan, 2022*). The role of PLAUR in the growth and progression of ccRCC has been demonstrated to be pivotal (*Wang et al., 2022a*). Researchers have discovered that PLAUR is a key gene signature within the immune system, carrying substantial importance for evaluating the effectiveness of immunotherapy and survival rates among ccRCC patients (*Gu et al., 2023*). Furthermore, the researchers suggest that an elevated expression of PLAUR is tightly linked to a poor prognosis in ccRCC. This gene not only serves as a crucial potential indicator for predicting macrophage infiltration and the immune microenvironment status, but also emerges as a promising target for ccRCC immunotherapy (*Wang et al., 2022b*). Based on these findings, we postulate that PLAUR can effectively serve as both a diagnostic and prognostic biomarker for ccRCC.

Despite the widespread recognition of the pivotal role played by PLAUR mRNA levels in ccRCC progression, the precise underlying mechanism remains elusive. The objective of this study was to elucidate the diagnostic significance of PLAUR mRNA levels in ccRCC and assess its potential as a prognostic marker. Furthermore, the mechanistic involvement of PLAUR in ccRCC development was explored through tissue-level analysis and *in-vitro* experiments.

## MATERIALS & METHODS

### Data acquisition

We obtained the RNAseq pan-cancer dataset in TPM format from UCSC XENA (https://xenabrowser.net/datapages/), which was uniformly standardized and included data from TCGA (https://portal.gdc.cancer.gov/) and GTEx. Additionally, we extracted the expression data of PLAUR (ENSG00000011422) gene for each sample and performed log2 transformation on the expression values. Furthermore, we excluded cancer types with less than three samples, resulting in a final dataset comprising 33 cancer species. The RNA-Seq expression data of PLAUR, along with corresponding clinical data of ccRCC patients and adjacent normal tissue, were also downloaded from TCGA database and GTEx databases. After excluding patients with missing clinical data, we obtained a total of 541 ccRCC patients and 72 adjacent normal tissues. The RNAseq data was converted to TPM format and standardized using log2 transformation. Moreover, the expression of PLAUR in ccRCC was further validated using GEO data sets GSE53757 (Platform: GPL570) and GSE15641 (Platform: GPL96). After excluding other types of tissue, we identified 72 ccRCC samples and 72 adjacent non-tumor samples from the GSE53757 dataset, while selecting 32 primary ccRCC samples and 23 normal kidney samples from the GSE15641

dataset. The raw data were downloaded as MINiML files. The extracted data underwent log2 transformation for normalization. Probes were converted to gene symbols based on the annotation information provided by the normalized data in the platform. The transcriptome of PLAUR and the associated clinical data were documented. Following that, the diagnostic significance of PLAUR content was evaluated as a distinctive indicator for ccRCC using receiver operating characteristic (ROC) curve analysis. Log-rank test was employed to assess the OS and progression-free interval (PFI) between patients with PLAUR levels. The accuracy of PLAUR hazard ratios (HR) was analyzed *via* Kaplan–Meier (K-M) curves and univariate Cox proportional hazards. A nomogram score and clinical profiles were used to predict the patient's outcomes. The independent predictive ability of the risk score in estimating prognosis profiles of ccRCC patients was investigated by conducting uni- and multivariate Cox regression analyses to examine the association between PLAUR transcript content and clinical characteristics. PLAUR mRNA content risk scores were then applied to generate outcome-related nomograms predicting the OS likelihoods among ccRCC patients. Finally, the nomograms were assessed for their predictive accuracy by measuring the concordance index (C-index) and examining the calibration curve.

### Enrichment analysis

The utilization of Gene Ontology (GO) evaluation offers a robust approach for classifying assessments of hepcidin-associated biological processes, cellular components, and molecular functions. Kyoto Encyclopedia of Genes and Genomes (KEGG) represents an experimental methodology that employs high-throughput technologies (*e.g.*, genome sequencing) to investigate advanced functionalities and biological systems. We investigated the functional mechanism of PLAUR action by performing GO and KEGG analysis using the R package cluster analyzer.

### Gene set enrichment analysis

The gene set enrichment analysis (GSEA) method was utilized to evaluate the association between predefined gene sets and the given phenotype by examining their distribution pattern in relation to genes linked with the phenotype. The gene sets with known functions (GO/MsigDB or other formats and the expression matrix) served as input. The entered gene expression levels were obtained by considering otypic associations and the newly formed gene sets were organized according to their functional annotations. The PLAUR expression level was used to categorize the samples into high expression groups ($\geq 50\%$) and low expression groups ($<50\%$), enabling the evaluation of phenotypic alterations.

### Pathway-level somatic alterations from Clinical Proteomic Tumor Analysis Consortium

Protein expression analysis information was supplied by the University of Alabama at Birmingham Cancer Data Analysis Portal (UALCAN) from Clinical Proteomic Tumor Analysis Consortium (CPTAC) datasets. CPTAC employed mass spectrometry to evaluate tumor biological measurements, quantify and identify proteins, and characterize the relevant protein groups in each tumor sample. This method systematically investigated

**Table 1 Clinical parameters of ccRCC patients with reducedand elevated PLAUR expression.**

| Parameters | Number of cases | PLAUR expression | | P-value |
| --- | --- | --- | --- | --- |
| | | Low (n = 7) | High (n = 13) | |
| Gender, n (%) | | | | |
| Male | 13 | 4 (20%) | 9 (45%) | 0.651 |
| Female | 7 | 3 (15%) | 4 (20%) | |
| Age, n (%) | | | | |
| ≤60 | 13 | 6 (30%) | 7 (35%) | 0.329 |
| >60 | 7 | 1 (5%) | 6 (30%) | |
| T.stage, n (%) | | | | |
| T1 | 9 | 6 (30%) | 3 (15%) | 0.017 |
| T2–T4 | 11 | 1 (5%) | 10 (50%) | |
| N.stage, n (%) | | | | |
| N0 | 13 | 6 (30%) | 9 (45%) | 0.613 |
| N1 | 7 | 1 (5%) | 4 (20%) | |
| M.stage, n (%) | | | | |
| M0 | 11 | 6 (30%) | 8 (40%) | 0.354 |
| M1 | 9 | 1 (5%) | 5 (25%) | |
| Pathologic stage, n (%) | | | | |
| Stage I | 9 | 6 (30%) | 3 (15%) | 0.017 |
| Stage II–IV | 11 | 1 (5%) | 10 (50%) | |

somatic changes at the pathway level, with a specific focus on crucial pathways and genes previously identified in various cancer types using domain expertise. This was achieved by integrating proteomic data, whole-exome sequencing, and copy number alteration information.

## Clinical samples

We collected tumor tissues and adjacent non-cancerous tissues from 20 ccRCC patients who underwent surgical resection at our institution. Patient clinical data, including gender, age, TNM stage, and Pathologic stage (I, II–IV), were meticulously documented (Table 1). The inclusion criteria were: ccRCC diagnosed in 2023 based on concurrent surgical resection, clinical information, laboratory examination, CT and MRI scans, and pathological evaluation, and complete medical records of the patients. This research received ethical approval from the Ethics Committee at institution of the Affiliated Hospital of Youjiang Medical University for Nationalities (No. YYFY-LL-2023-052). The participants were enrolled only after writing the informed consent.

## Immunohistochemistry staining

PLAUR expression level in tissues was assessed by immunohistochemistry (IHC) staining. After fixing the specimen tissue with 4% paraformaldehyde and embedding it in paraffin, 4 µm slices were made. The slices were dewaxed, antigen-retrieved, and then incubated overnight at 4 °C with PLAUR-targeting antibodies (Zenbio, Durham, NC, USA). Wash off the primary antibody with PBS, incubate with secondary antibody at room temperature

**Table 2  The sequences for PLAUR-specific siRNAs.**

| siRNA | Sense | Antisense |
|---|---|---|
| siPLAUR-1 | 5′-CACUCAGAGAAGACCAACATT-3′ | 5′-UGUUGGUCUUCUCUGAGUGTT-3′ |
| siPLAUR-2 | 5′-CGAGGUUGUGUGUGGGUUATT-3′ | 5′-UAACCCACACACAACCUCGTT-3′ |
| siPLAUR-3 | 5′-GCCAGUGUUACAGCUGCAATT-3′ | 5′-UUGCAGCUGUAACACUGGCTT-3′ |
| NT siRNA | 5′-UUCUCCGAACGUGUCACGUTT-3′ | 5′-ACGTGACTGTTCGCATCAT-3′ |

for 30 min, and then perform diaminobenzidine staining. The slides were photographed using a Nikon fluorescence microscope (Tokyo, Japan).

## Cell culture

RCC cell lines (786-O, 769-P, ACHN, and Caki-1) and human embryonic kidney cells (HEK293) were obtained from ProCell Corporation (Wuhan, China). HEK293 and ACHN cell lines were cultured in Minimum Essential Medium (Oricell, Shanghai, China), Caki-1 cells were cultured in McCoy's 5A medium (Oricell, Shanghai, China), and 786-O and 769-P cells were cultured in RPMI-1640 medium (Oricell, Shanghai, China). All the basal culture medium were enhanced with 10% fetal bovine serum (Oricell, Shanghai, China) and 1% penicillin/streptomycin (Solarbio, Beijing, China). The cells were maintained (37 °C, 5% $CO_2$) under humidified conditions.

## Establishment of cell lines with PLAUR knockdown using siRNA construction

Transient transfection of PLAUR siRNAs (siPLAUR-1, siPLAUR-2, and siPLAUR-3) and the non-targeting siRNA control (NT siRNA) was performed provided by HanBio (HanBio, Shanghai, China). An equal number of well-cultured ccRCC cells (786-O, ACHN) were seeded in six-well plates with complete culture medium. When the cells reached 70% ∼80% confluence, each well was treated with 200 µL of a mixed reagent containing 5 µL of Lipofectamine 3000 transfection reagent (Thermo Fisher Scientific, Waltham, MA, USA). After a 6 h incubation period, the transfection process was completed. To confirm successful transfection and assess efficiency, Fluorescently labelled siRNA was used. The sequences for PLAUR-specific siRNAs and NT siRNA are listed in Table 2.

## RNA extraction and qRT-PCR

RT-qPCR was employed to assess the expression of PLAUR mRNA in ccRCC and adjacent tissues. The total RNA was isolated from cell lines or 50 mg tissue samples (preserved at −80 °C) using the TRIzol reagent (Thermo Fisher Scientific, Waltham, MA, USA) protocol. The quality of the RNA was verified by employing a NanoDrop One spectrophotometer (Thermo Fisher Scientific, Waltham, MA, USA), ensuring an A260/280 ratio falling within the range of 1.9–2.1. The RNA was subjected to DNA digestion and reverse transcription using the ToloScript All-in-one RT EasyMix (TOLOBIO, China). Afterwards, 1.0 µg of RNA was converted into cDNA utilizing ToloScript Reverse TransCripatase in a 20 µL reaction mixture. The reverse transcription process took place at 50 °C for 15 min followed by a brief incubation at 85 °C for 5 s. The resulting cDNA was stored at −80 °C. Subsequently, qPCR analysis was performed employing a LightCycler 96 Real-Time PCR

**Table 3  The primer sequences utilized.**

| Target | sense | antisense |
|---|---|---|
| PLAUR | 5′-CGAGGTTGTGTGTGGGTTAGACT-3′ | 5′-CAGGCACTGTTCTTCAGGGCT-3′ |
| GAPDH | 5′-CAGGAGGCATTGCTGATGAT-3′ | 5′-GAAGGCTGGGGCTCATTT-3′ |

Detection System with a reaction mixture volume of 20 µL consisting of 2 µL of cDNA, 10 µL of 2 × PowerUp SYBR Green Master Mix (Thermo Fisher Scientific, Waltham, MA, USA), and gene-specific primers (PLAUR, GAPDH). The PCR program involved the activation of a UDG enzyme at 50 °C for 120 s, followed by predegeneration at 95 °C for 120 s. This was succeeded by 40 cycles of denaturation at 95 °C for 15 s and annealing at 60 °C for another 15 s. An extension step was then performed at a temperature of 70 °C for one minute. For melting curve analysis, the samples were subjected to temperatures of 95 °C for 15 s, followed by a decrease to 60 °C maintained for one minute, and finally raised back up to 95 °C again for an additional duration of fifteen seconds. The resulting dissolution curve exhibited a single peak. Gene-specific primers (listed in Table 3) were designed to amplify fragments of approximately 100 bp, which were verified by conducting a BLAST search on the NCBI database. The fold changes in RNA abundance were determined using the $2^{(-\Delta\Delta CT)}$ method, with GAPDH utilized as an internal reference. The slope values ranged from $-3.39$ to $3.1$, while maintaining an $R^2$ value of $\geq 0.98$. To establish the limit of detection, we employed LOD = 2.5 for targeting molecules and calculated a confidence interval of 95%. Data analysis eliminated technical replicates that exhibited substantial deviations from the other two values among three technical replicates. Each group consisted of 3–20 biological replicates, and every sample underwent three technical replicates to ensure reliable outcomes.

## Western blotting

After cell lysis, the total protein was extracted through centrifugation (4 °C, 14,000 g, 10 min) and subsequently separated by SDS-PAGE gel electrophoresis (EpiZyme, Cambridge, MA, USA). Protein samples were then transferred onto PVDF (Merck Millipore, Rahway, NJ, USA) membranes. The membranes were subsequently blocked with 10% skim milk at room temperature for 60 min, followed by overnight incubation with a primary antibody on a shaker at 4 °C After washing off the primary antibody with PBS, they were subsequently incubated with anti-rabbit secondary antibody (1:7500 diluted, Proteintech, Rosemont, IL, USA) for 1 h at room temperature. The protein bands were imaged using the AI600 imaging system (GE, USA) following addition of enhanced chemiluminescence reagent kits (EpiZyme, Cambridge, MA, USA). The quantification and analysis of protein densitometry were performed using Image J software v1.8.0. The antibodies utilized in the experiments are listed in Table S1.

## Cell counting kit-8 assay

Cell counting kit-8 (CCK-8) assay (UElandy, China) was used to assess cell proliferation ability. Cells transfected with NT siRNA and siPLAUR were seeded onto 96-well plates for incubation at 37 °C for 1, 2, 3, and 4 days. Subsequently, 10 µL of CCK-8 reagent was

added to each well and incubated at 37 °C in the absence of light for a duration of 2 h. Ultimately, the absorbance was measured at a wavelength of 450 nm.

## Transwell assay

The migration and invasion of cells were evaluated using 24-well plates equipped with Trans chambers (Biofil, China). The upper chamber was added with 60 μL Matrigel (Invitrogen, Waltham, MA, USA) (Matrigel coating is essential for Transwell invasion assays, while not required for Transwell migration assays). The cells were resuspended in RPMI 1640 medium without fetal bovine serum and then seeded at a concentration of $2 \times 10^4$ cells (in a volume of 200 μL) into the upper chamber of the Transwell apparatus. Subsequently, the lower chamber was supplemented with 500 μL of RPMI 1640 culture containing 20% fetal bovine serum. Incubation was carried out at a temperature of 37 °C with a $CO_2$ level maintained at 5% for a duration of 24 h. Following incubation, the cells were fixed using a solution consisting of paraformaldehyde (4%) and stained with crystal violet solution (0.1%), followed by thorough washing with water. The invasive cells were quantified by examining five randomly selected fields.

## Flow cytometry

According to the instructions of the Cell Cycle Staining Kit (Multi Sciences, Shanghai, China), $1 \times 10^6$ cells were collected and washed with PBS. The supernatant was then centrifuged. Subsequently, 1 ml of DNA Staining solution and 10 μl of Permeabilization solution were added, followed by vortex oscillation for 5–10 s. The cells were incubated at room temperature for 30 min. Flow cytometry was performed using the lowest loading speed available. Furthermore, early and late apoptotic cells were analyzed using the Annexin V-FITC / PI apoptosis kit (Multi Sciences, Shanghai, China). Cells from the cell medium were aspirated and covered with an appropriate amount of thawed Accutase. After leaving them at room temperature for 5 min until they began to shrink, they were gently blown away. Washed by centrifugation with pre-cold PBS, a collection of $1 \times 10^5$ cells (including those in the culture supernatant) was obtained. The collected cells were resuspended in Binding Buffer diluted in double steaming and mixed with 5 μl Annexin V-FITC and 10 μl PI per tube. Gently vortexing them together, they were then incubated for 5 min at room temperature. Finally, on a flow cytometer machine (Thermo Fisher Scientific, USA), Annexin V-FITC (Ex = 488 nm; Em = 530 nm) was measured through its respective channel while PI fluorescence passed through another channel (Ex = 535 nm; Em = 615 nm).

## Statistical analysis

The statistical analysis was conducted utilizing both SPSS22.0 and R software (V4.2.1) for data processing. Differential significance analysis was performed using the unpaired Wilcoxon rank sum and signed rank tests. ROC curve generation and PLAUR threshold value computation were performed using the pROC software package. The potential association between the patient's clinicopathological profile and PLAUR content was assessed through the Kruskal and chi-square tests. The survival curves were generated

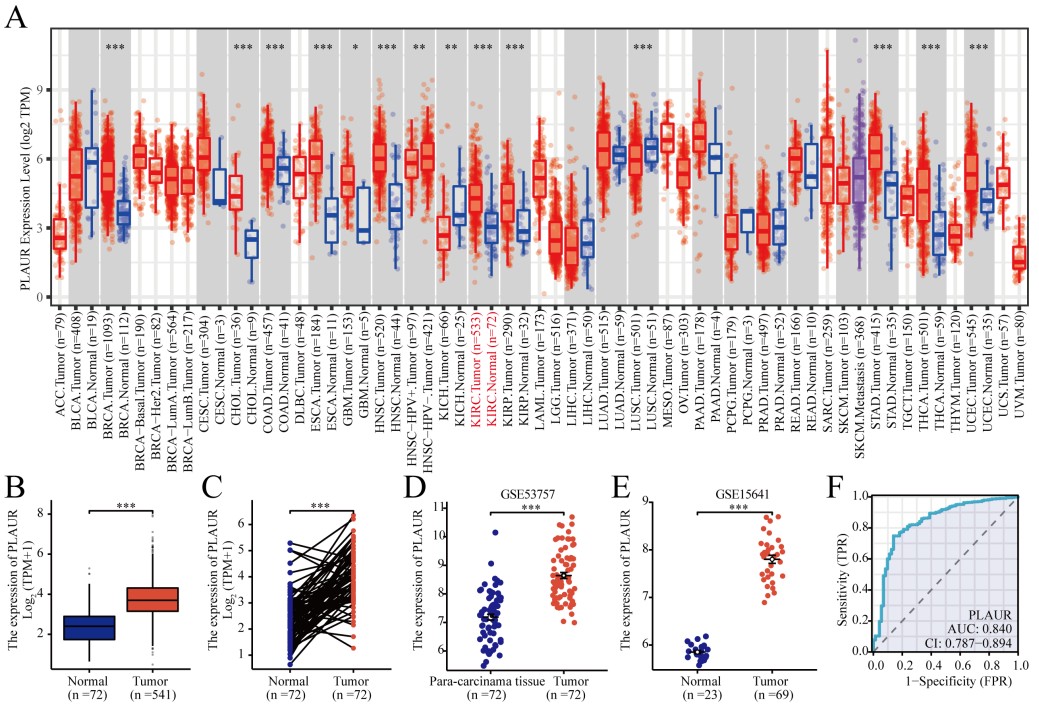

**Figure 1  Expression of PLAUR mRNA in pancarcinoma.** (A) The expression of PLAUR in different tumor types retrieved from the TCGA database was demonstrated. (B) The expression of PLAUR was analyzed in 541 cases of ccRCC and 72 normal tissues.(C) PLAUR expression in ccRCC ($n = 72$) and the matched adjacent normal tissues ($n = 72$) was analyzed. PLAUR expression in ccRCC was analyzed using the GSE53757 (D) and GSE15641 (E) datasets obtained from the GEO database. (F) The ROC curve for PLAUR in ccRCC was graphed. * $P < 0.05$, ** $P < 0.01$, *** $P < 0.001$.

using K-M analysis. Graphical work was carried out using R software and GraphPad Prism 9.

## RESULTS

### The public database reveals a significant upregulation of PLAUR expression in ccRCC

According to TCGA database analysis, PLAUR exhibited significantly higher expression in 12 out of the 33 analyzed cancer types (Fig. 1A). The expression level of PLAUR mRNA was significantly elevated in ccRCC tumor tissues compared to normal tissues ($P < 0.001$, Fig. 1B). Additionally, paired data analysis results demonstrated a significant increase in PLAUR mRNA expression in ccRCC tumor tissues relative to the adjacent healthy tissues ($P < 0.001$, Fig. 1C). Moreover, datasets (GSE53757 and GSE15641) were used for further validation. It was also found that PLAUR expression was notably higher in ccRCC than in normal tissues ($P < 0.001$, Figs. 1D and 1E). The area under the curve (AUC) was determined as 0.840, which suggested that PLAUR strongly indicated the presence of ccRCC at an optimal threshold value of 3.164 (Fig. 1F). These results revealed that PLAUR holds promise as a biomarker for ccRCC progression.

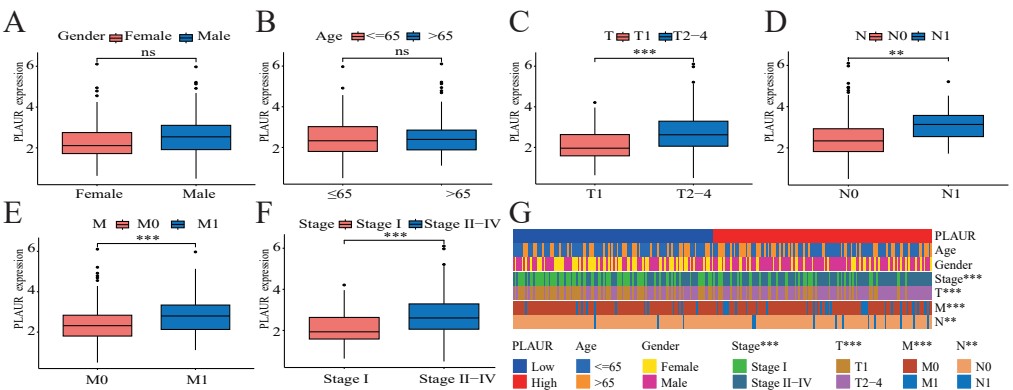

**Figure 2** **The correlation between PLAUR mRNA expression and clinical features is being analyzed based on the TCGA database.** The correlation of PLAUR mRNA expression in ccRCC with gender (A), age (B), tumor stage T (C), lymph node metastasis stage N (D), distant metastasis stage M (E), and pathologic stage (F) was investigated. (G) The correlation between PLAUR expression and the patient's clinical characteristics in ccRCC was displayed using a heat map. ** $P < 0.01$, *** $P < 0.001$, *ns*: not significant.

## The pathological features of ccRCC patients correlated significantly with the expression level of PLAUR mRNA

The correlation between PLAUR expression and the clinical pathological features of patients was evaluated by analyzing data obtained from the TCGA database. Based on the findings of the analysis, there was no significant correlation between PLAUR mRNA expression and gender ($P = 0.622$, Fig. 2A) or age ($P = 0.57$, Fig. 2B), while there was a strong association between PLAUR mRNA expression and tumor stage T ($P < 0.001$, Fig. 2C), stage N ($P = 0.0147$, Fig. 2D), stage M ($P < 0.001$, Fig. 2E), and pathological stage ($P < 0.001$, Fig. 2F). These clinical correlations were presented using a heat map in Fig. 2G.

## PLAUR expression was associated with unfavorable outcomes in ccRCC

The clinical data profiles of ccRCC patients are presented in Table 4. K-M analysis was used to evaluate the prognostic. Patients with elevated PLAUR expression had remarkably reduced OS ($HR = 1.77$, $P < 0.001$) and PFI ($HR = 1.78$, $P < 0.001$) compared to those with reduced PLAUR expression (Figs. 3A and 3B). The expression of PLAUR mRNA exhibited a robust correlation with both survival time and survival status in the TCGA dataset (Fig. 3C). Furthermore, a multivariate Cox regression analysis was conducted to further evaluate the predictive significance of PLAUR on clinical outcomes. Based on the results of the multivariate analysis, it was observed that PLAUR mRNA expression independently served as a prognostic factor for overall survival (OS) in ccRCC ($HR = 1.722$, 95% CI [1.466–2.023], $P < 0.001$). Furthermore, age ($HR = 1.032$, 95% CI [1.017–1.047], $P < 0.001$), grade ($HR = 2$. CI [1.854–2.780], $P < 0.001$), and stage ($HR = 1.857$, 95% CI [1.629–2.116], $P < 0.001$) were also discovered as independent prognostic factors for OS in ccRCC patients (Figs. 3D and 3E). Subsequently, an OS estimation nomogram model was developed by integrating PLAUR mRNA content with other clinicological profiles obtained

**Table 4 Clinical parameters of ccRCC patients with reduced and elevated PLAUR expression form the TCGA database.**

| Parameters | Number of cases | PLAUR expression | | P-value |
|---|---|---|---|---|
| | | Low (*n* = 270) | High (*n* = 271) | |
| Gender, n (%) | | | | |
| Female | 187 | 96 (17.7%) | 91 (16.8%) | 0.629 |
| Male | 354 | 174 (32.2%) | 180 (33.3%) | |
| Age (years), n (%) | | | | |
| <= 60 | 269 | 130 (24%) | 139 (25.7%) | 0.465 |
| >60 | 272 | 140 (25.9%) | 132 (24.4%) | |
| T stage, n (%) | | | | |
| T1 | 279 | 160 (29.6%) | 119 (22%) | <0.001 |
| T2-4 | 262 | 110 (20.3%) | 152 (28.1%) | |
| N stage, n (%) | | | | |
| N0 | 242 | 122 (47.3%) | 120 (46.5%) | 0.014 |
| N1 | 16 | 3 (1.2%) | 13 (5%) | |
| M stage, n (%) | | | | |
| M0 | 429 | 228 (44.9%) | 201 (39.6%) | <0.001 |
| M1 | 79 | 26 (5.1%) | 53 (10.4%) | |
| Pathologic stage, n (%) | | | | |
| Stage I | 273 | 158 (29.4%) | 115 (21.4%) | <0.001 |
| Stage II–IV | 265 | 110 (20.4%) | 155 (28.8%) | |
| Histologic grade, n (%) | | | | |
| G1 | 14 | 7 (1.3%) | 7 (1.3%) | 0.960 |
| G2-4 | 519 | 256 (48%) | 263 (49.3%) | |

from the TCGA dataset (Fig. 3F). Notably, our calibration curves demonstrated excellent accuracy of the newly generated in predicting OS among ccRCC patients (Fig. 3G).

## Functional enrichment analysis showed the functional pathway of PLAUR in ccRCC

To investigate the potential mechanisms underlying tumor progression mediated by PLAUR, we performed a comparative analysis of gene expression between samples with high and low levels of PLAUR expression. The expression profiles (HTSeq-TPM) of the high and low PLAUR mRNA expression groups were compared using an wilcoxon rank sum test within the limma Package software, without accounting for any covariates or random effects in the models. The differentially expressed genes (DEGs) were identified using a threshold of |log2Fold Change|>1.5 and adjusted $P < 0.001$. A total of 658 differentially expressed genes (DEGs) were detected, consisting of 586 upregulated genes and 72 downregulated genes. According to the selection criteria, we selected the top 10 DEGs for validation and observed their significant differential expression between ccRCC and normal samples (Fig. S1). To gain deeper understanding of the functional implications of all these DEGs in ccRCC, we conducted GO and KEGG functional enrichment analyses using the clusterProfiler package. The enriched biological processes included immunoglobulin production, production of molecular mediators of immune

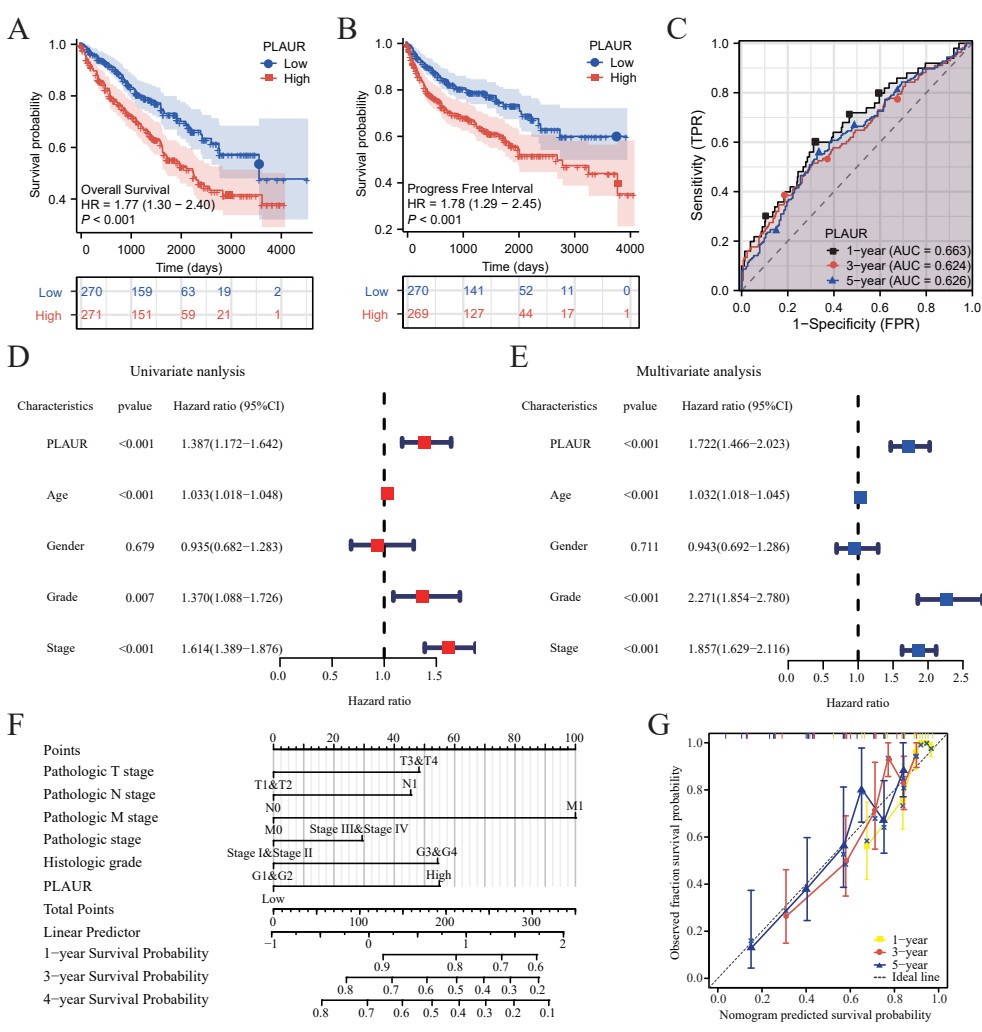

**Figure 3 The prognostic significance of PLAUR levels in ccRCC was analyzed.** (A) OS survival curves from the TCGA database ($n = 541$). (B) PFI survival curves from the TCGA database ($n = 539$). (C) Time-dependent PLAUR ROC curve analysis in ccRCC. (D) Forest plot illustrating the results of univariate Cox regression analysis in ccRCC. (E) The forest plot presents a visual representation of the results obtained from conducting a multivariate Cox regression analysis on ccRCC data. (F) A nomogram integrating PLAUR and other prognostic factors in ccRCC from the TCGA database. (G) Nomogram's calibration curve.

response, and antimicrobial humoral response. The cellular component analysis included immunoglobulin complex, external side of plasma membrane, and blood microparticle. Molecular function analysis included antigen binding activity, receptor ligand activity, and cytokine activity (Figs. 4A and 4C). Additionally, KEGG pathway enrichment analysis results revealed significant enrichments in cytokine–cytokine receptor interaction, IL-17 signaling pathway, viral protein interaction with cytokines and cytokine receptors, rheumatoid arthritis, chemokine signaling pathway, and JAK-STAT signaling pathway (Fig. 4B). Furthermore, GSEA analysis results identified key pathways associated with

PLAUR expression. With the criteria of *FDR* < 0.25 and *P* < 0.05, the significantly enriched pathways were PI3K-AKT-mTOR-VITD3, cell cycle, and apoptosis signaling pathway (Fig. 4D). We standardized the TCGA-KIRC data downloaded and curated from the TCGA database. Pearson correlation analysis was performed between PLAUR gene and all other molecules, with *p*-value correction using the Benjamini–Hochberg method. We filtered for factors with |*Cor*| > 0.5 & *P* < 0.05, resulting in 283 strongly correlated factors. These factors were subjected to biological pathway enrichment analysis using FunRich 3.1.3 software, revealing that PLAUR exhibited significant co-expression with genes enriched in key signaling pathways such as the mTOR signaling pathway and class PI3K signaling events mediated by AKT signaling pathway (Fig. 4E). Additionally, pathway-level somatic alterations from CPTAC datasets revealed an association between PLAUR expression and mTOR pathway dysregulation (Fig. 4F).

### The expression of PLAUR was found to be significantly increased in ccRCC

To investigate potential disparities in PLAUR expression in ccRCC, we collected and performed IHC analysis on 20 pairs of ccRCC tissues and their corresponding adjacent tissues As shown by our findings, PLAUR expression in the ccRCC tissues was noticeably elevated compared to that in the corresponding adjacent tissues (*P* = 0.0026, Fig. 5A). Next, the association between PLAUR expression and pathological characteristics of RCC was analyzed. A total of 20 ccRCC patients were categorized into the high PLAUR (*n* = 13) and low PLAUR (*n* = 7) groups based on the median level of PLAUR. Our findings demonstrated a noteworthy association between PLAUR levels and T stage (*P* = 0.017), as well as pathologic stage (*P* = 0.017) in RCC patients. However, no significant correlations with gender (*P* = 0.651), age (*P* = 0.329), N stage of lymphatic metastasis (*P* = 0.613), or M stage of distant metastasis (*P* = 0.354) among these patients were observed (Table 1). Additionally, the qRT-PCR results unequivocally validated a significant upregulation of PLAUR mRNA expression in 20 RCC samples when compared to that in the corresponding adjacent tissues (most of the time, *P* < 0.001, Fig. 5B). In this study, limited by the small number of clinical samples, only 20 tumor and the adjacent tissues of ccRCC patients were collected. The results would be more convincing if the sample size could be increased. The levels of PLAUR mRNA and protein expression were assessed in four different ccRCC cell lines (Caki-1, ACHN, 786-O, and 769-P) as well as in the human embryonic kidney cell line HEK293 were evaluated using qRT-PCR and western blot analysis. Our findings consistently demonstrated a significant upregulation of PLAUR expression in ccRCC cells compared to its expression in HEK293 cells (*P* < 0.001, Figs. 5C and 5D).

### Knockdown of PLAUR hampered the proliferation, migration, and invasion of ccRCC cells

For investigating the impact of PLAUR expression on ccRCC progression, 3 siRNA fragments targeting different positions were transfected into 786-O and ACHN cells. Subsequently, both mRNA and protein levels of PLAUR were significantly reduced in the knockdown cells compared to that in controls (all *P* < 0.001, Figs. 6A and 6B). Furthermore, according to CCK-8 assay results, there was a notable decrease in cell proliferation among

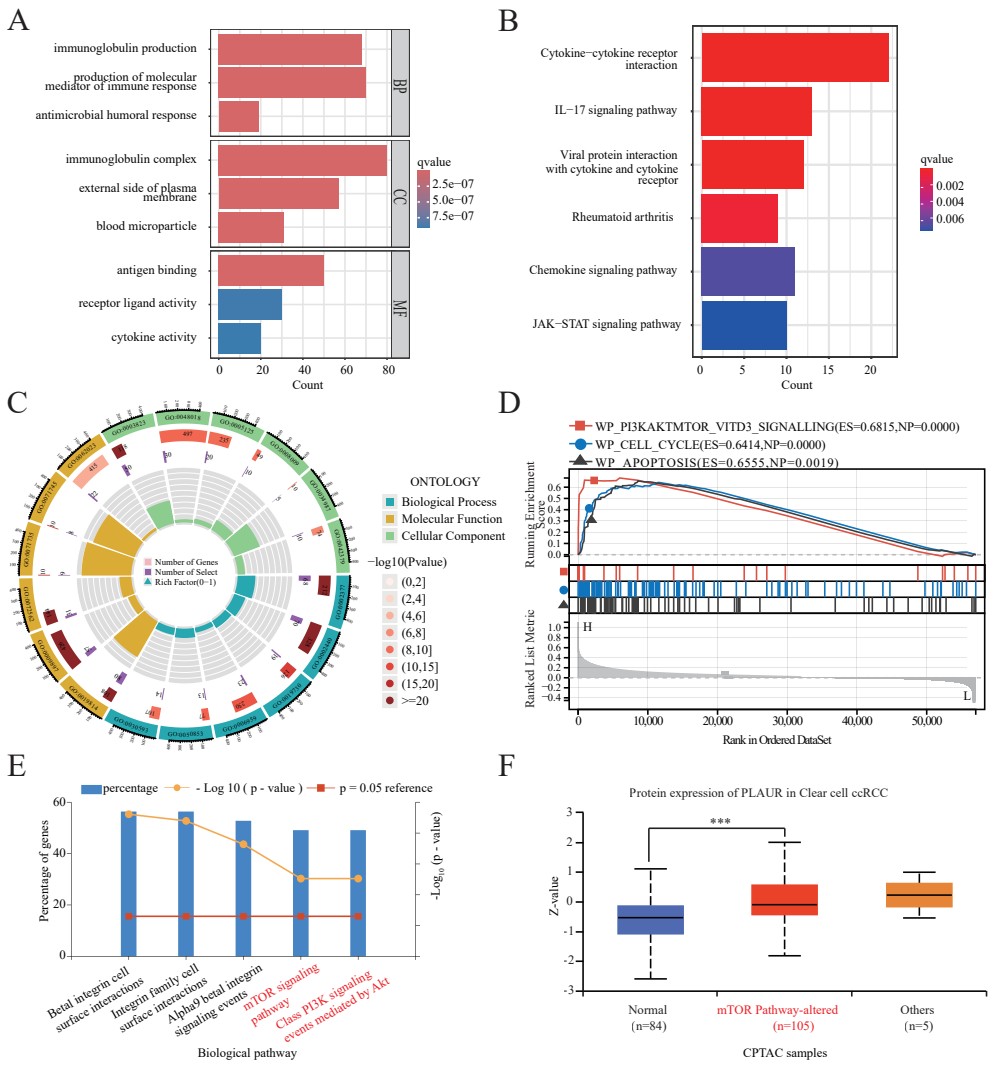

**Figure 4  PLAUR functional enrichment in ccRCC was analyzed.** (A) GO and (B) KEGG analyses of DEGs in reduced and elevated PLAUR expression samples. (C) Circle plot for GO functional enrichment analysis. (D) GSEA functional analysis of genes enriched in representative networks. (E) Signal pathway enrichment analysis of PLAUR coexpression genes. (F) PLAUR proteomic expression profile based on mTOR pathway status in ccRCC from the CPTAC database. *** $P < 0.001$.

the knockdown group as compared to the NT siRNA group for both 786-O and ACHN cells (all $P < 0.001$, Fig. 6C). Additionally, Transwell assays revealed that the migratory and invasive capabilities of cells were significantly impaired upon PLAUR knockdown (all $P < 0.01$, Fig. 6D). The findings indicated that the proliferation, migration, and invasion of 786-O and ACHN cells were significantly suppressed upon PLAUR knockdown.

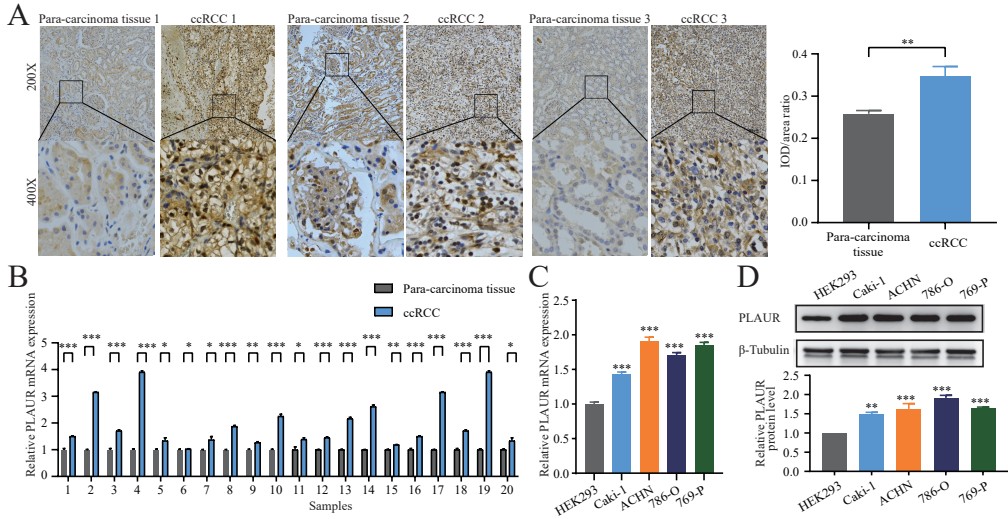

**Figure 5** **The expression of PLAUR was increased in ccRCC tissues and cell lines.** (A) The relative protein level of PLAUR in 20 pairs of ccRCC tissues and the adjacent tissues was examined by IHC; IOD/area ratio of the indicated IHC images results confirmed that PLAUR level was upregulated in ccRCC tissues. qRT-PCR was used to detect the relative mRNA expression of PLAUR in 20 pairs of ccRCC tissues and the adjacent tissues (B) and human embryonic kidney cells HEK293 and 4 ccRCC cells (C). (D) Western blot was used to detect the expression of PLAUR protein in human embryonic kidney cells HEK293 and 4 ccRCC cells. * $P < 0.05$, ** $P < 0.01$, *** $P < 0.001$.

## Knockdown of PLAUR induced cellular apoptosis, modulated the cell cycle, and inhibited epithelial-mesenchymal transition (EMT) in ccRCC

Functional enrichment analysis using GSEA revealed that there was a significant association of PLAUR with cell cycle and apoptosis (Fig. 4D). Hence, flow cytometry analysis was performed to experimentally validate this finding. The results revealed a significant increase in early, late, and total apoptosis levels in the PLAUR knockdown group compared to NT siRNA group cells ($P < 0.001$, Fig. 7A). Moreover, knockdown of PLAUR significantly increased the proportion of cells in the G2 phase, while decreasing it in the G1 phase, indicating cell cycle arrest at the G2 phase in ccRCC cells ($P < 0.001$, Fig. 7B). EMT has been generally considered to be related to tumor cell invasion and metastasis. In this study, Western blot was used to evaluate the impact of PLAUR on the expression of EMT-related genes in ccRCC cells. Notably, downregulation of PLAUR significantly reduced levels of Vimentin, N-cadherin, and MMP9 while simultaneously increasing the level of E-cadherin (all $P < 0.05$, Fig. 7C). Collectively, PLAUR exerted regulatory effects on ccRCC cell progression by suppressing apoptosis, modulating cell cycle progression, and inducing EMT.

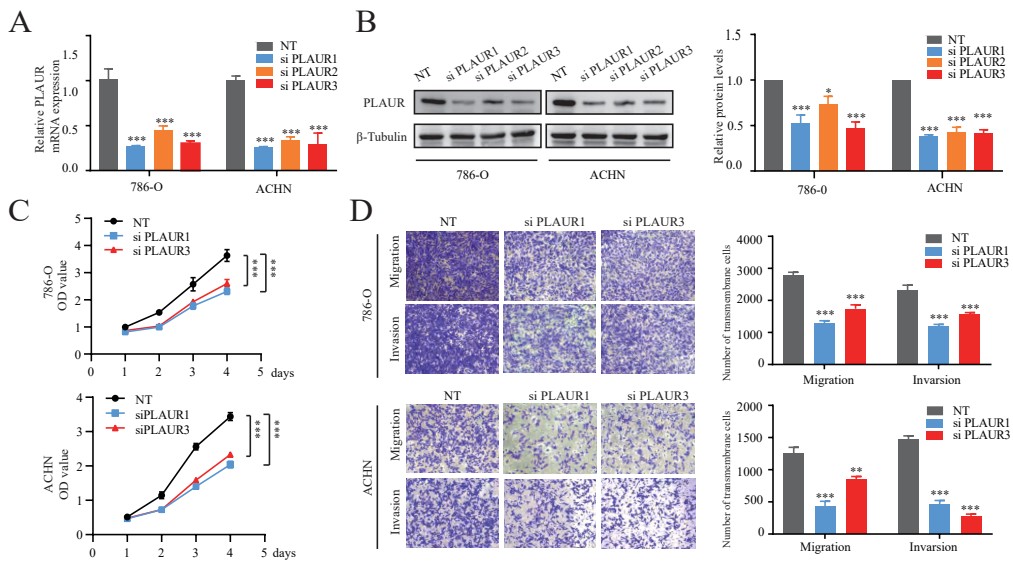

**Figure 6** **The knockdown of PLAUR expression hampered the *in vitro* malignant progression ability of ccRCC cells.** The relative expression of PLAUR in 786-O and ACHN cells was detected using qRT-PCR (A) and western blotting (B). (C) The CCK-8 assay was employed to evaluate the impact of PLAUR knockdown on the proliferatio of 786-O and ACHN cells. (D) Transwell migration and invasion assay was used to detect the effect of PLAUR knockdown on the migration and invasion of 786-O and ACHN cells.* $P < 0.05$, ** $P < 0.01$, *** $P < 0.001$.

## PLAUR facilitated ccRCC proliferation, migration, and invasion by modulating the PI3K/AKT/mTOR signaling pathway

Based on the enrichment analysis results, it was found that PLAUR exerted its function in ccRCC through modulation of the PI3K/AKT/mTOR signaling pathway (Figs. 4D–4F). Based on the findings from western blot analysis, PLAUR knockdown significantly influenced the activation of the PI3K/AKT/mTOR pathway, leading to a decrease in the phosphorylation levels of PI3K, AKT, and mTOR. Additionally, downstream expression of HIF-$\alpha$ and MMP9 was decreased by this pathway, which could be reversed by the AKT activator (SC-79; ab146428, Abcam, Cambridge, UK) ($P < 0.05$, Fig. 8A). Additionally, The proliferation, migration, and invasion abilities of ACHN cells in the NT siRNA, siPLAUR, SC-79, and siPLAUR + SC-79 groups were evaluated using CCK-8 and Transwell assays. Our results demonstrated that SC-79 could reverse the inhibitory effect of siPLAUR on ACHN proliferation, migration, and invasion ($P < 0.05$, Figs. 8B and 8C). Taken together, these findings suggested that the PLAUR/PI3K/AKT/mTOR axis played a critical role in regulating EMT as well as the proliferation, migration, and invasion of ccRCC cells.

## DISCUSSION

ccRCC is a predominant and extremely aggressive subtype of RCC, making up around 70 to 80 percent of all RCC cases (*Jiang et al., 2020*). Even though ccRCC patients with early and localized lesions have a 5-year OS rate exceeding 90%, those with distant

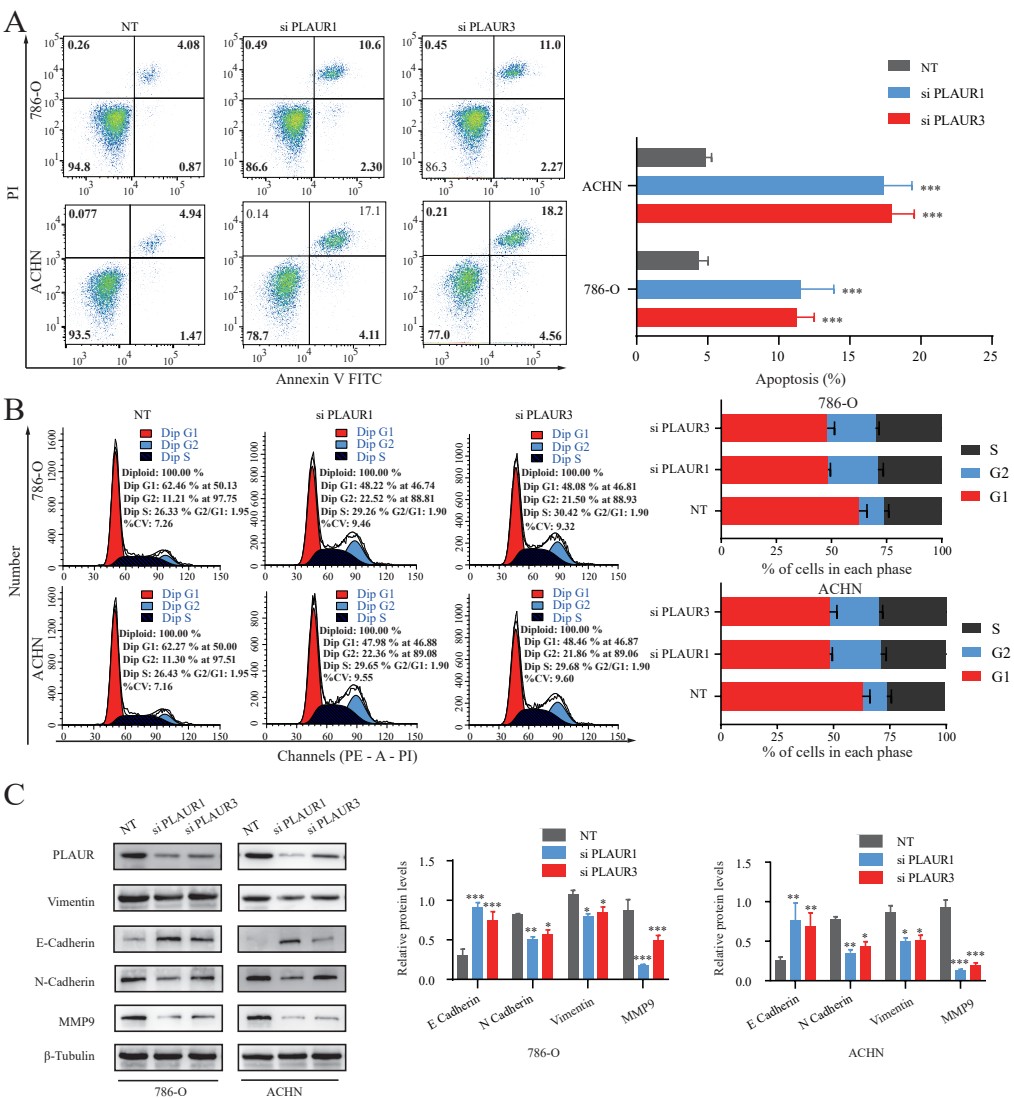

**Figure 7** **The effects of PLAUR on ccRCC cell cycle progression, apoptosis, and EMT were analyzed.**
(A) The apoptosis was assessed using flow cytometry with the Annexin V-FITC/PI flow double staining kit, and the population of cells positive for Annexin V was designated as undergoing apoptosis. PLAUR knockdown promoted cell apoptosis in ccRCC cells. (B) The effect of PLAUR knockdown on the cell cycle in 786-O and ACHN cells was analyzed, ccRCC cells were regulated and blocked in G2 phase. (C) The protein levels of vimentin, N-cadherin, E-cadherin, and MMP9 in 786-O and ACHN cells following PLAUR knockdown were assessed using the Western blot. * $P < 0.05$, ** $P < 0.01$, *** $P < 0.001$.

metastases have a dramatically low OS rate as low as 12% (*Wang et al., 2020b*). Due to the lack of specific clinical symptoms, ccRCC metastasis results in delayed diagnosis, limited treatment options, and poor response to standard chemo- and radiotherapies, with a 5-year disease-free survival rate of only 12% (*Wang et al., 2020a*). ccRCC is mainly manifested by lumbar and abdominal masses, gross hematuria, and fatigue. Partial or radical nephrectomy serves as the standard intervention (*Zou et al., 2019*). However, the

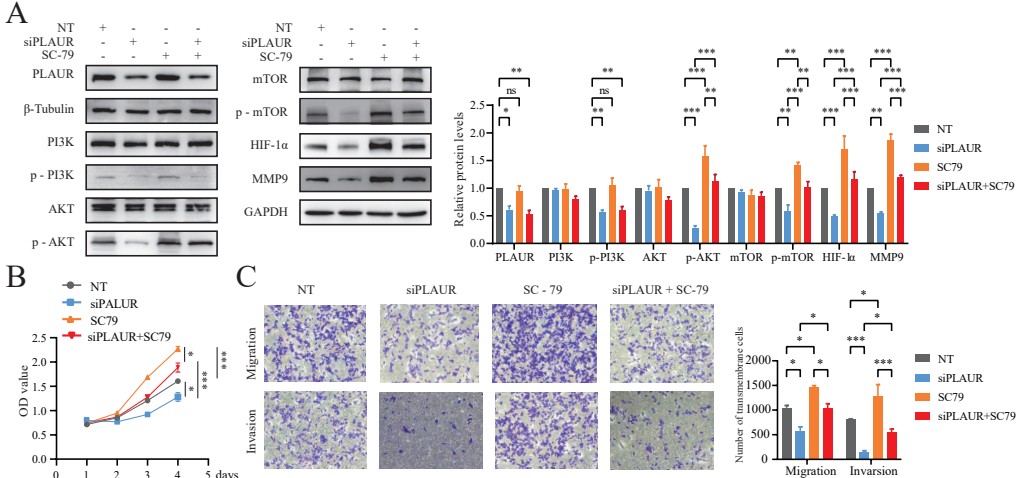

**Figure 8 PLAUR accelerated ccRCC proliferation, migration, and invasion *via* the PI3K/AKT/mTOR pathway.** (A) The Western blotting analysis confirmed the presence of LPLAUR, PI3K, p-PI3K, AKT, p-AKT, mTOR, p-mTOR, HIF-$\alpha$, and MMP9 in the siNC group and the siPLAUR group. Additionally, these proteins were also detected in the SC-79 (AKT activator) group and the siPLAUR + SC-79 group. (B) The proliferation ability of ccRCC cells in different groups was assessed using the CCK-8 assay. (C) The migration and invasion ability of ccRCC cells in different groups was assessed using the Transwell as-say. *$P < 0.05$, **$P < 0.01$, ***$P < 0.001$, *ns*: not significant.

advanced ccRCC still exhibits an extremely low OS rate (*Sidaway, 2023*). Nevertheless, its underlying pathogenesis remains elusive. Several studies have indicated that the progression of ccRCC could potentially be influenced by genetic factors, abnormalities in chromosomes, and occurrences of gene fusion events (*Lasorsa et al., 2023*). Herefore, it is imperative to identify biomarkers that can accurately prognosticate prognosis of ccRCC and possess potential therapeutic targets.

Members of the fibrinolytic system play a pivotal role in various biological processes, encompassing cell of cytokines and growth factors, immune response and angiogenesis, as well as tumor cell invasion, migration and other phenomena (*Zhang et al., 2022*). Under normal physiological conditions, stromal cells secrete a complex collagen fiber protein network that effectively impedes tumor cell migration. However, the activation of the fibrinolytic system network by tumor cells leads to the establishment of a conducive environment for tumor advancement (*Mahmood & Rabbani, 2021*). Accumulating studies have revealed the oncogenic significance of PLAUR. For example, recent studies have elucidated its pivotal role in the proliferation, invasion, and metastasis processes of esophageal cancer (*Xie et al., 2023*), breast cancer (*Zhang et al., 2023*), bladder urothelial carcinoma (*Liu et al., 2021*), and prostate cancer (*Kimura et al., 2020*). *Zhang et al. (2022)* have demonstrated *in vitro* and *in vivo* that PLAUR promotes the malignant process of gastric cancer by affecting the cell cycle, apoptosis, and EMT. Therefore, PLAUR is an important regulator of tumor cell migration. Currently, the role of PLAUR in ccRCC remains unelucidated. Preliminary studies have found that PLAUR is significantly higher in ccRCC tissues, and the increased expression of PLAUR is tightly implicated in the

poor prognosis of patient (*Wang et al., 2022a*; *Wang et al., 2022b*). In the field of medicine, the search for effective therapeutic targets has always been a primary focus of scientists in treating various diseases. Recently, PLAUR, an important cell surface receptor, has garnered increasing attention from researchers and is considered a potential therapeutic target (*Killeen et al., 2008*; *Li & Cozzi, 2007*). The investigators also discovered that PLAUR facilitates the development of resistance to sunitinib in ccRCC (*Yang et al., 2024*). Given its significant role in multiple diseases, researchers have begun exploring therapies that target PLAUR by inhibiting or modulating its function (*Mazar, 2001*; *Zhou et al., 2018*). Currently, several drugs targeting PLAUR have entered clinical trials with preliminary results demonstrating some efficacy (*Nozaki et al., 2006*; *Tyndall et al., 2008*). However, despite its promising potential as a therapeutic target, PLAUR still faces numerous challenges including understanding its specific mechanisms of action in different diseases and overcoming issues related to drug safety and effectiveness during development.

In this study, the TCGA database was downloaded and bioinformatics investigation was conducted to evaluate the expression profile of PLAUR. Our findings revealed a high expression of PLAUR in various tumor tissues. Among them, PLAUR expression was also significantly higher in ccRCC tissues than in matched normal kidney tissues. These results were further verified through qRT-PCR and IHC clinical tissue samples. Moreover, the ROC curve analysis demonstrated that PLAUR held potential as a diagnostic biomarker for accurately distinguishing between ccRCC and normal tissues. Additionally, PLAUR mRNA expression was significantly upregulated in advanced TNM and pathological stage patients with higher histological, indicating its close association with ccRCC malignancy. These results were consistent with previous investigations conducted by *Ye et al. (2022)*. Furthermore, a robust association was observed between elevated levels of PLAUR and unfavorable overall survival (OS) outcomes in patients, as substantiated by the analysis of Kaplan–Meier curves and univariate analysis findings. Importantly, multivariate analysis results confirmed that elevated PLAUR levels served as an independent risk factor for patient outcomes. The progn estimation nomogram also supported the strong correlation between PLAUR levels and patient outcomes. Collectively, these findings highlighted the potential of PLAUR as a superior biomarker candidate for identifying poor prognosis in ccRCC patients.

Furthermore, GO and KEGG analyses using the TCGA database revealed that PLAUR is primarily associated with the biological function of an immune response. The co-expressed genes with PLAUR were predominantly enriched in pathways such asytokine-cytokine receptor interaction. In recent years, mounting evidence has demonstrated the crucial role of immune cell infiltration in tumor progression regulation, immunotherapy response control, and clinical efficacy. On this basis, cancer immunotherapy has achieved significant advancements and breakthroughs, making it one of the most promising strategies for treating ccRCC (*Barata & Rini, 2017*). To date, numerous well-established mechanisms of immunosuppression have been proposed to enhance the anti-tumor effects by increasing various immunosuppressive cells and molecules while reducing tumor antigen levels. In recent years, a significant association between PLAUR expression and various indicators of immune suppression, such as immune checkpoints and cytokines, has

been discovered by *Zeng et al. (2021)*, with the correlation also being validated in patient samples. Additionally, *Wang et al. (2022b)* analyzed the impact of PLAUR on the tumor immune microenvironment and found that it plays a crucial role in ccRCC progression. Therefore, we will further explore the relationship between PLAUR and the progression of ccRCC.

PLAUR has been identified as a pivotal gene in promoting tumorigenesis, representing a potential novel target for cancer (*Ahn et al., 2019*). According to the literature, silencing of PLAUR notably impedes the proliferation and migration of pancreatic adenocarcinoma cells, while inducing cell apoptosis *via* inhibiting the ERK/p38 signaling pathways (*Xue et al., 2009*). *Semina et al. (2020)* have observed that PLAUR knockdown could reduce neuroblastoma Neuro2a cell proliferation *in vitro*, highlighting the critical role of PLAUR expression in maintaining the epithelial phenotype in Neuro2a cells. Through our bioinformatic analysis and experimental validation, we have identified that PLAUR potentially facilitates the progression of ccRCC *via* activation of the PI3K/AKT/mTOR pathway. The PI3K/AKT/mTOR pathway is a complex signaling network involved in regulating crucial cellular processes such as growth, proliferation, and apoptosis. Normally, this pathway maintains normal cellular development and metabolism; however, aberrations within its components can lead to excessive cell proliferation and ultimately trigger carcinogenesis (*Glaviano et al., 2023*). The Pi3k/Akt/mTOR pathway is frequently aberrantly activated in clear cell renal cell carcinoma, which can be attributed to genetic mutations, protein expression abnormalities, and other contributing factors. This dysregulated activation of the pathway facilitates cancer cell proliferation and metastasis, thereby expediting disease progression (*Badoiu et al., 2023*). Extensive research has been conducted on anti-cancer drugs to suppress the aberrant activation of the Pi3k/Akt/mTOR pathway, with several drugs targeting this pathway currently undergoing clinical trials and demonstrating promising therapeutic effects. These agents effectively inhibit cancer cell growth and metastasis by attenuating pathway activity (*Asati, Mahapatra & Bharti, 2016*). In addition to pharmacotherapy, researchers are investigating alternative strategies targeting the Pi3k/Akt/mTOR pathway for combating cancer. For instance, novel therapeutic modalities such as gene therapy and immunotherapy are progressively emerging, offering promising prospects in the management of ccRCC. These approaches involve direct modulation of pivotal molecules within the pathway or activation of the body's innate immune system to combat malignancy (*Xu et al., 2020*). In our study, PLAUR knockdown effectively attenuated the proliferation, migration, invasion, and EMT of ccRCC cells, along with the simultaneous inhibition of the cell cycle and apoptotic progression. The association between PLAUR and cancer progression was further investigated. *In vitro* experiments was conducted to evaluate the impact of PLAUR knockdown on the PI3K/AKT/mTOR pathway and its downstream targets HIF-1$\alpha$ and MMP9. The latest research findings have demonstrated that heightened levels of HIF-1$\alpha$ are indicative of tumor progression and associated with an unfavorable prognosis (*Masoud & Li, 2015*). Our findings revealed a reduction in AKT and mTOR phosphorylation without altering total protein levels. Moreover, decreased HIF-1$\alpha$ and MMP9 expressions were observed, which could be reversed by the AKT pathway activator SC79. These findings indicate that PLAUR

may facilitate ccRCC progression through activation of the PI3K/AKT/mTOR pathway, thereby confirming its role in tumor advancement. However, certain limitations exist in this study. A larger number of clinical samples is required to establish the clinicopathological relevance effectively. Moreover, our investigation on the downregulation of PLAUR inhibiting the PI3K/AKT/mTOR pathway lacks a direct association between PLAUR and this pathway when employing only a single AKT activator (SC-79). For future research, inclusion of specific PI3K and mTOR inhibitors for experimental purposes will provide more compelling evidence. Additionally, in order to complement the findings of our *in vitro* studies, future research endeavors will include corresponding *in vivo* investigations assessing the impact of PLAUR knockdown on animal models of ccRCC. Furthermore, efforts will be made to establish a stronger clinicopathological correlation.

## CONCLUSIONS

In conclusion, PLAUR exhibited a high expression level in ccRCC, which can serve as an independent prognostic marker indicating a poor prognosis in ccRCC. The proliferation, migration, and invasion of 786-O and ACHN cells were remarkably suppressed upon PLAUR knockdown. Additionally, PLAUR knockdown effectively induced cellular apoptosis, modulated the cell cycle, inhibited the EMT process, and attenuated the activation of the PI3K/AKT/mTOR signaling pathway, which may represent a key mechanism underlying ccRCC progression.

### Funding

This work was supported by the 2020 High-level Talent Research Project of the Affiliated Hospital of Youjinag Medical University for Nationalities (No. Y202011717) and the Health Commission of Guangxi Autonomous Region self-funded research project (No. Z-L20230905). The funders had no role in study design, data collection and analysis, decision to publish, or preparation of the manuscript.

### Grant Disclosures

The following grant information was disclosed by the authors:
2020 High-level Talent Research Project of the Affiliated Hospital of Youjinag Medical University for Nationalities: Y202011717.
Health Commission of Guangxi Autonomous Region self-funded research project: Z-L20230905.

### Competing Interests

The authors declare there are no competing interests.

### Author Contributions

- Tianzi Qin conceived and designed the experiments, performed the experiments, analyzed the data, prepared figures and/or tables, authored or reviewed drafts of the article, and approved the final draft.

- Minyu Huang performed the experiments, prepared figures and/or tables, and approved the final draft.
- Wenjuan Wei performed the experiments, prepared figures and/or tables, and approved the final draft.
- Wei Zhou conceived and designed the experiments, performed the experiments, prepared figures and/or tables, and approved the final draft.
- Qianli Tang conceived and designed the experiments, performed the experiments, analyzed the data, authored or reviewed drafts of the article, and approved the final draft.
- Qun Huang analyzed the data, prepared figures and/or tables, and approved the final draft.
- Ning Tang performed the experiments, analyzed the data, prepared figures and/or tables, and approved the final draft.
- Shasha Gai performed the experiments, prepared figures and/or tables, and approved the final draft.

## Human Ethics

The following information was supplied relating to ethical approvals (*i.e.*, approving body and any reference numbers):

The Affiliated Hospital of Youjiang Medical University for Nationalities granted Ethical approval to carry out the study within its facilities (Ethical Application Ref: YYFY-LL-2023-052).

## Data Availability

The raw data is available at figshare: Qin, Tianzi (2024). Raw Data. figshare. Dataset. https://doi.org/10.6084/m9.figshare.25271203.v4.

The flow cytometry data is available at FlowRepository, ID: FR-FCM-Z784.

http://flowrepository.org/id/FR-FCM-Z784

The raw data is available in the Supplemental Files.

## Supplemental Information

Supplemental information for this article can be found online at http://dx.doi.org/10.7717/peerj.17555#supplemental-information.

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
