# Peer review of "PLAUR facilitates the progression of clear cell renal cell carcinoma by activating the PI3K/AKT/mTOR signaling pathway"

_PeerJ, doi:10.7717/peerj.17555_

## Round 0.1 · original submission · Minor Revisions

Reviewers provided valuable insights and offered impactful feedback on this manuscript. Kindly incorporate their suggestions and carefully address all comments in your manuscript. Thank you.

Reviewer 1 ·

Basic reporting

The manuscript would benefit from a more comprehensive discussion of the potential clinical implications of the findings. The authors should elaborate on how PLAUR could be utilized as a diagnostic or prognostic biomarker in ccRCC and discuss any potential limitations of their study

Experimental design

The authors should provide more details on the patient cohort used for the IHC and qRT-PCR validation experiments. Information such as sample size, inclusion/exclusion criteria, and patient demographics should be clearly stated in the methods section.

The in vitro experiments lack appropriate controls. The authors should include a non-targeting siRNA control to ensure the specificity of the PLAUR knockdown effects observed . Additionally, a rescue experiment where PLAUR is overexpressed in the knockdown cells would strengthen the conclusions drawn from the functional assays.

Validity of the findings

The authors claim that PLAUR knockdown inhibits the PI3K/AKT/mTOR pathway based on the western blot results. However, the use of a single AKT activator (SC-79) is insufficient to establish a direct link between PLAUR and this pathway. Additional experiments using specific PI3K and mTOR inhibitors would provide more convincing evidence.

Additional comments

No comments.

Reviewer 2 ·

Basic reporting

In the paper “PLAUR facilitates the progression of clear cell renal cell carcinoma by activating the PI3K/AKT/mTOR signaling pathway”, the authors demonstrated the associations between PLAUR and ccRCC progression and identified the associations between PLAUR expression and PI3K/AKT/mTOR signaling pathway through bioinformatics and in vitro studies. The paper provided sufficient details in background, the importance of the study and the hypothesis to test.

Experimental design

While the findings from the paper are interesting and inspiring, there are a few suggestions to improve on clarifications.
1. Please describe the samples from TCGA database, GSE53757 and GSE15641 datasets and cite the datasets. Is there any quality control performed on the datasets?
2. What is the sample distribution for PLAUR mRNA and protein expression?
3. In section 3.4, would the DEGs and pathways identified here also be differentially expressed between the ccRCC and normal samples?
4. In section 3.4, please clarify the pathways were enriched with up or down regulated DEGs.
5. Please clarify how the DEGs were identified (i.e. which statistical model was used) and whether any covariates/random effects were controlled for in the models.
6. Please clarify on how the gene co-expression network analysis was performed (LINE 310-313)

Validity of the findings

From the bioinformatics analysis and in vitro studies, the paper demonstrated associations between PLAUR and ccRCC progression and knocking down PLAUR led to cell apoptosis, etc. However, associations do not equal causality. End points from in vivo studies with ccRCC knock down or knock out may be needed to confirm the causal roles of PLAUR and PI3K/AKT/mTOR signaling pathway to ccRCC progression. Please discuss on this.

Additional comments

Minor comments:
1. Please use the same color to demonstrate the same experimental groups in figures 6-8.
2. Please revise the titles for the sub sections in the Results section. The titles should summarize the main findings for the sub sections.

Reviewer 3 ·

Basic reporting

The language used in the manuscript is clear and professional. The authors have provided sufficient background information to understand the context of the research. The abstract and introduction sections give a detailed background of the PLAUR gene, its association with various cancers, and the significance of studying its role in ccRCC.

Experimental design

The methods are described in detail, allowing for reproducibility of the study. The use of TCGA and GTEx databases for data acquisition, the employment of various assays, and the statistical analysis are well-documented. The research question is well-defined, relevant, and meaningful. The study aims to investigate the relationship between PLAUR and ccRCC and its potential mechanism in promoting tumor progression. The experimental design is appropriate to address the research questions posed by the authors. The use of both bioinformatic analysis and in vitro experiments provides a comprehensive approach to the study.

Validity of the findings

The findings that PLAUR is upregulated in ccRCC and associated with poorer prognosis are impactful for the understanding of ccRCC progression. The study also adds to the current knowledge by demonstrating the potential mechanism through which PLAUR may promote tumor progression via the PI3K/AKT/mTOR signaling pathway. The data appears to be robust and controlled, with the use of multiple techniques to validate findings, such as qRT-PCR, IHC staining, Western blot analysis, and various cellular assays.

Additional comments

Overall, the manuscript is well-written and presents a thorough investigation into the role of PLAUR in ccRCC. However, there are certain areas that could be improved.

1) The authors should consider increasing the number of clinical samples to strengthen the statistical power.
2)To complement the in vitro findings, in vivo studies could be conducted to observe the effects of PLAUR knockdown in animal models of ccRCC.of their findings and to better establish clinicopathological relevance.
3) Investigating the long-term effects of PLAUR knockdown on ccRCC progression could provide additional information on the potential of PLAUR as a therapeutic target.
4) Correlating the expression of PLAUR with patient responses to existing treatments could help in understanding its potential role in personalized medicine for ccRCC.

---

## Round 0.2 · accepted · Accept

The authors have conducted excellent work on this manuscript, incorporating feedback from reviewers to enhance the clarity and depth of the content. They have successfully elucidated the role of PLAUR in the clear cell renal carcinoma, providing a detailed and comprehensive explanation of this complex biological process. This manuscript now offers a clearer and more insightful understanding of the mechanisms at play, thanks to the thoughtful integration of the reviewers' suggestions.

Reviewer 1 ·

Basic reporting

The authors have resolved all my concerns.

Experimental design

The authors have resolved all my concerns.

Validity of the findings

The authors have resolved all my concerns.

Additional comments

The authors have resolved all my concerns.

Reviewer 2 ·

Basic reporting

The paper is well written with sufficient background information and literature references.

Experimental design

The study used rigorous approaches and methods to study the research question. All the comments from previous review were comprehensively addressed.

Validity of the findings

The conclusions are well stated and supported by the results.

Reviewer 3 ·

Basic reporting

No comments

Experimental design

No comments

Validity of the findings

No comments

Additional comments

The authors have addressed most of my queries and concerns and have made the necessary updates to ensure data transparency. Therefore, I am ready to endorse the manuscript for publication.